# The impact of macroeconomic policy uncertainty on micro-enterprise investment efficiency—Empirical evidence from China

**Bing Zhou**[1,2], **Bo Liang**[1] *, **Licheng Bie**[1]

**1** Chengdu-Chongqing Region Double City Economic Circle Construction Research Institute, Chongqing Technology and Business University, Chongqing, China, **2** School of Accounting, Chongqing Technology and Business University, Chongqing, China

* 617556326@qq.com

**Data Availability Statement:** All relevant data are within the article and its Supporting Information files.

**Funding:** Funder Name: Chongqing Social Sciences Association Project Grant Number:

## Abstract

The impact of macroeconomic policy uncertainty (EPU) on micro-level entities has garnered increasing attention in economic circles. This study examines the influence of EPU on the efficiency of investments made by China's A-share listed companies between 2016 and 2021. Using a panel fixed effect model for analysis, the research reveals that EPU has a notable adverse effect on the investment efficiency of enterprises. Furthermore, it suggests that advancements in digital finance, strong ESG performance, and enhanced entrepreneurial confidence can mitigate this negative impact effectively. The study also highlights that enterprises with lower valuation, shareholder control, limited audit reputation, and no bank connections are more vulnerable to the impact of EPU on investment efficiency compared to those with higher valuation, manager control, strong audit reputation, and bank connections. Consequently, future efforts should be directed towards enhancing the stability and relevance of economic policies, promoting digital finance, and enhancing corporate governance structures.

## 1. Introduction

Uncertainty has long been a significant challenge within the economic system, impacting the formulation of government policies. To address this issue, it is crucial to select a suitable research subject for analysis. China, with its socialist market economic system, exhibits distinct policy-oriented characteristics. The economic policies crafted by the Chinese government wield a profound influence on the country's enterprises. Given China's status as the world's leader in listed companies, it offers a robust sample size for research, enhancing the credibility and applicability of the study. As such, China serves as an ideal research subject. Since the beginning of the 21st century, China's economic policy uncertainty index has experienced peaks during key events such as China's accession to the WTO in 2001, the SARS outbreak in 2003, the global financial crisis in 2008, the US sovereign credit rating decline and European debt crisis in 2011, the stock issuance registration system reform and stock market crash in 2015, the introduction of the fuse mechanism in 2016, and the new coronavirus epidemic

2019WT42 Grant Recipient:Bing Zhou Funder Name: Key Research Platform Open Project of Chongqing Technology and Business University Grant Number:KFJJ2022043 Grant Recipient:Bing Zhou In our study, the funders played a pivotal yet arms-length role, primarily functioning as oversight entities ensuring the project adheres to the agreed-upon objectives and ethical standards. Their involvement was strategically confined to the realms of financial support and high-level guidance, thereby preserving the research team's autonomy in terms of methodology design, data collection, analysis, and interpretation. Regular progress updates were shared with sponsors, enabling them to monitor the project's trajectory without influencing the scientific process. This approach aligns with international guidelines on research sponsorship, which emphasize the importance of maintaining independence between funding sources and research outcomes to uphold scientific credibility. Additionally, a clear conflict of interest policy was in place, which all sponsors and researchers adhered to, further safeguarding the integrity of our findings. Through this, we were able to leverage the resources provided by our sponsors effectively while preserving the objectivity and authenticity of our research endeavor.

**Competing interests:** The authors declare no competing interests.

outbreak in 2020. Analyzing the impact of macroeconomic policy uncertainty on micro-level behavior and its underlying mechanisms can offer valuable insights for China in formulating stable macroeconomic policies and fostering high-quality economic development.

According to existing literature, scholars studying the impact of economic policy uncertainty on the social economy focus on the relationship between economic policy uncertainty and macro factors such as economic development, the foreign exchange market, and financial risk. Meanwhile, scholars studying the efficiency of enterprise investment pay more attention to the impact of corporate governance, the external institutional environment, and the informal system on investment efficiency [1–3]. Only a few scholars have integrated economic policy uncertainty and enterprise investment efficiency into the same research framework. However, there are significant differences in the research conclusions of these scholars, which can be broadly categorized into two views. There are two theories regarding the impact of economic policy uncertainty on enterprise investment. The first is the *constraint theory*, which suggests that uncertainty will worsen underinvestment, leading to reduced investment efficiency. The second is the *efficient decision-making theory*, which posits that uncertainty will increase the prudence and rationality of investment decisions, thereby improving investment efficiency. Furthermore, the limited literature available fails to systematically examine the mechanism and heterogeneity of economic policy uncertainty that affects corporate investment efficiency. Therefore, our research holds theoretical significance and importance in three key aspects. Our research contributes to resolving differences in existing literature by providing new evidence supporting the *constraint theory*. Secondly, this study analyses the impact of economic policy uncertainty on enterprise investment efficiency through three factors: digital financial development, enterprise ESG performance, and entrepreneur information. The findings provide theoretical support and practical measures for enterprises to cope with economic policy uncertainty and enhance investment efficiency. Thirdly, this study identifies the varying impact of economic policy uncertainty on different types of enterprises based on their valuation level, governance structure, audit reputation, and bank-enterprise relationship. This information is valuable for enterprises to manage the uncertain impact of economic policy. The above outlines the theoretical significance and practical application of this study. Given the significant number of micro-enterprises in China, which are heavily impacted by macro policies, addressing the aforementioned issues is of utmost importance.

## 2. Theoretical analysis and research hypothesis

In theory, enterprises often only need to consider the economic factors of the market when making investment decisions [4]. However, in the real world, the investment efficiency of enterprises is not only affected by economic factors. Non-economic factors such as information asymmetry, agency costs, financing constraints, and managers' self-confidence often also cause inefficient investment [5, 6]. As an important exogenous variable faced by enterprises, EPU will have a significant impact on the investment efficiency of enterprises. First of all, EPU will aggravate the complexity of investment environment, and it is difficult for enterprise decision makers to objectively predict the economic situation [7], so as to misunderstand or misjudge the prospect of investment projects (cash flow, return period, etc.), so as to make inefficient investment. Secondly, EPU will reduce the quality of accounting information. Enterprises need to fully grasp the business situation and financial situation of enterprises to make investment decisions. The existence of uncertainty (tax policy, accounting standards) makes it impossible for business managers to use high-quality accounting information to make effective investment decisions [8]. Finally, economic policy uncertainty will increase corporate credit constraints and financing difficulties. The long-standing *ownership*

*discrimination* in China's credit market will lead to distortions in the allocation of credit resources [9]. In order to alleviate this credit constraint, enterprises often obtain related loans by holding bank equity, which will increase the debt leverage of enterprises, resulting in the imbalance of enterprise asset structure and the decline of investment efficiency. Based on the above analysis, we puts forward the hypothesis:

H1: Economic policy uncertainty (EPU) has a negative impact on corporate investment efficiency.

Through empirical research, a large number of literatures have found that digital finance plays an important role in the investment behavior of micro-subjects: from the perspective of structural effect, digital finance promotes the investment efficiency of enterprises, and is significant in breadth and depth [10]. From the perspective of the path of action, financing constraints play an intermediary effect, which is more obvious in the under-investment sample [11, 12]. In addition, from the perspective of technological progress, the development of digital finance can drive the R&D investment of enterprises, achieve effective deleveraging, and improve the level of financial robustness of enterprises [13], thus reducing their own inefficient investment behavior. As mentioned above, economic policy uncertainty will reduce the credit availability of enterprises and increase the financing constraints of enterprises, resulting in inefficient investment of enterprises. Then, can the attribute of digital finance to alleviate financing constraints play a role in the process of EPU affecting the investment efficiency of enterprises? Accordingly, this paper proposes the following hypothesis:

H2: The improvement of the development level of digital finance will weaken the negative impact of EPU on corporate investment efficiency.

ESG refers to the generalized responsible investment strategy of the economy from the three aspects of environment, social responsibility and corporate governance. With the increasing openness of China's capital market, the participation rate of foreign capital and institutional investors is rising. They have a high degree of recognition of ESG investment concept, forcing domestic enterprises and investors to pay more attention to ESG performance [14]. Enterprise ESG performance will mainly affect the investment efficiency of enterprises through two aspects: one is to reduce agency costs. Strong ESG performance is a benign signal transmitted by enterprises to the outside, which is conducive to improving corporate reputation and value, effectively enhancing the trust of principal-agent parties, reducing agency costs, and reducing the emergence of inefficient investment conditions [15]. Second, ease credit constraints. Strong ESG performance discloses more firm-specific information to creditors such as banks, and also meets the expectations of governments and regulators, helping companies to obtain external financial support and alleviate underinvestment. Accordingly, this paper proposes the following hypothesis:

H3: Strong ESG performance will weaken the negative impact of EPU on corporate investment efficiency.

Frequently changing macroeconomic policies will increase the level of EPU, resulting in micro-enterprises in a fuzzy, poor-expected operating environment [16]. This will seriously affect the confidence of entrepreneurs, and thus affect the investment efficiency of enterprises: First, the increase in EPU will hinder the accumulation of physical capital and human capital [17]. Reduce the expected return of enterprises, reduce the confidence of entrepreneurs, and make them adopt more prudent investment decisions. Micro-enterprises may therefore miss better investment opportunities, resulting in a decline in investment efficiency. Secondly, high EPU will also impact managers' confidence by exacerbating

information loss or error [18]. In addition, in the case of high EPU, internal decision-making of enterprises is often more inclined to the strategy of *cash is king*. The superposition of the two aggravates the underinvestment of enterprises. The negative impact of EPU may be transmitted from entrepreneurial confidence to corporate investment efficiency, so will improving entrepreneurial confidence weaken this negative impact? Therefore, this paper puts forward the hypothesis:

H4: The improvement of entrepreneurial confidence will weaken the negative impact of EPU on corporate investment efficiency.

## 3. Model design and variable description

### 3.1. Benchmark model design

In order to explore the impact of EPU on corporate investment efficiency and verify Hypothesis H1, this paper constructs the following benchmark model:

$$Abs_{it} = \beta_0 + \beta_1 Epu_t + \beta_2 X_{it} + \eta_i + \varepsilon_{it} \tag{1}$$

Where the subscripts $i$ and $t$ denote company and year, respectively. The explained variable $Abs_{it}$ represents the inefficient investment level of the enterprise. $EPU_{it}$ represents economic policy uncertainty, and parameter $\beta_1$ measures the impact of economic policy uncertainty on corporate inefficient investment. $X_{it}$ represents a series of control variables, $\beta_2$ is the influence coefficient corresponding to the control variable, and $\eta_i$ represents the individual fixed effect. In order to ensure that the regression results are more robust, we use the clustering robust standard error regression model to estimate.

In order to further investigate the mechanism of digital financial development, enterprise ESG performance and entrepreneur confidence in the process of EPU affecting enterprise investment efficiency, and verify hypotheses H2, H3 and H4, this paper introduces the interaction term of mechanism variable M and explanatory variable EPU on the basis of Model (1), and constructs the following moderating effect model:

$$Abs_{it} = \mu_0 + \mu_1 Epu_{it} + \mu_2 M_{it} + \mu_3 Epu_{it}*M_{it} + \mu_4 X_{it} + \eta_i + \varepsilon_{it} \tag{2}$$

Where the subscripts $i$ and $t$ denote company and year, respectively. The explained variable $Abs_{it}$ represents the inefficient investment level of the enterprise. $Epu_{it}$ represents economic policy uncertainty, and parameter $\mu_1$ measures the impact of economic policy uncertainty on corporate inefficient investment. $M_{it}$ represents three moderating variables, including digital finance development (DFI), enterprise ESG performance (ESG) and entrepreneur confidence (Con). $\mu_2$ is the influence coefficient corresponding to the moderating variable. $Epu_t*M_{it}$ is the intersection of the explanatory variable $Epu_t$ and the mechanism variable $M_{it}$. $\mu_3$ is the influence coefficient corresponding to the multiplicative term. $X_{it}$ represents a series of control variables, $\mu_4$ is the influence coefficient corresponding to the control variable, and $\eta_i$ $\eta_i$ represents the individual fixed effect. To ensure that the regression results are more robust, we use the clustering robust standard error regression model to estimate.

### 3.2. Variable declaration

Explained variable: enterprise inefficient investment (*Abs*). Because the investment efficiency of enterprises is not easy to measure directly, the non-efficiency investment level of enterprises is usually used as the reverse index of investment efficiency of enterprises. Referring to Richardson (2006)'s measurement method of inefficient investment [19], we use model residuals to

measure the level of inefficient investment of enterprises, and constructs the following regression model.

$$Inv_{i,t} = \theta_0 + \theta_1 Lev_{i,t-1} + \theta_2 Growth_{i,t-1} + \theta_3 Age_{i,t-1} + \theta_4 Cash_{i,t-1} + \theta_5 Return_{i,t-1} + \theta_6 Size_{i,t-1}$$
$$+ \theta_7 Inv_{i,t-1} + Industry_i + \varepsilon_{i,t} \qquad (3)$$

Where the $Inv_{it}$ represents the new investment expenditure of the enterprise in the current period, $Lev_{i,t-1}$ represents the company's asset-liability ratio in the previous period; $Growth_{i,t-1}$ indicates the growth rate of the company's operating income in the previous period; $Age_{i,t-1}$ is the listing year of the previous period; $Cash_{i,t-1}$ is the cash ratio of the enterprise in the previous period, which is equal to the ratio of cash holdings to total assets at the beginning of the period; $Return_{i,t-1}$ is the stock return of the previous period; $Size_{i,t-1}$ is the size of the enterprise, expressed as the logarithm of total assets; $Inv_{i,t-1}$ is the new investment expenditure of the enterprise in the previous period; $Industry_i$ represents the industry dummy variable. Next, the regression of Model (3) is carried out, and the absolute value of its residual is taken as the proxy variable of inefficient investment. The larger the absolute value is, the more serious the inefficient investment phenomenon is.

Explanatory variables: economic policy uncertainty (EPU). In relevant research, economic policy uncertainty is typically measured using indicators such as changes in government [20], political territory [21], and the Chinese EPU index compiled by Baker et al. [22] using the *South China Morning Post*. Davis et al. [23] developed the Chinese economic policy uncertainty index using data from the Chinese mainland newspapers *People's Daily* and *Guang Ming Daily*. The economic policy uncertainty index of mainland China was developed by Huang & Luk P. based on ten newspapers [24], including *Beijing Youth Daily*, *Guangzhou Daily*, *Liberation Daily*, *People's Daily Overseas Edition*, *Shanghai Morning Post*, *Southern Metropolis Daily*, *Beijing News*, *Today's Evening News*, *Wenhui Daily*, and *Yangcheng Evening News*. While the first and second indicators are exogenous to corporate investment, they lack continuity and time variability. The third and fourth indicators may not have sufficient explanatory power for economic policy uncertainty in mainland China due to limitations in newspaper selection. Considering the aforementioned factors, this paper has chosen the economic policy uncertainty index of mainland China, developed by Huang Y. and Luk P. [24], as the proxy variable for economic policy uncertainty (EPU). The index is measured as follows:

Huang Y. & Luk [24] count the number of occurrences of articles discussing economic policy uncertainty in leading daily general interest Chinese-language newspapers. Huang Y. & Luk P. [24] construct a monthly index starting in January 2000 by searching for relevant keywords in the electronic archives of ten newspapers: *Beijing Youth Daily*, *Guangzhou Daily*, *Jiefang Daily*, *People's Daily Overseas Edition*, *Shanghai Morning Post*, *Southern Metropolis Daily*, *The Beijing News*, *Today Evening Post*, *Wen Hui Daily* and *Yangcheng Evening News*. Huang Y. & Luk P. obtain newspaper contents and search for related keywords in the digital archives wisers Information Portal. This platform covers important and influential papers from large cities representative of the newspaper market in urban areas. For each newspaper, they search for articles which contain at least one keyword in each of the three criteria, namely (1) Economy, (2) Uncertainty, and (3) Policy. Table 1 shows the keywords in each criterion and their English translation.

In order to match the EPU index to the year level, we refer to the practice of Li et al. [6], the annual data of the year is obtained by dividing the arithmetic average of the twelve monthly data of the year by 100. In terms of robustness test, the fiscal policy uncertainty index (FEPU) and monetary policy uncertainty index (MEPU) of mainland China developed by Huang & Luk P. [24] are selected.

**Table 1. Relevant keywords for compiling China economic policy uncertainty index.**

| Criteria | English |
|---|---|
| (1) Economic | Economic/economy/financial |
| (2) Uncertainty | Uncertainty/uncertain |
| | Volatile |
| | Unstable/unclear |
| | Unpredictable |
| (3) Policy | Policy/measures |
| | Politics |
| | Government/authority |
| | President |
| | Prime minister |
| | Reform |
| | Regulation |
| | Fiscal |
| | Tax |
| | People's Bank of China/PBOC |
| | Deficit |
| | Interest rate |

**The control variables.** Refer to the study of Gu & Zhu [8], this paper mainly selects micro-enterprise characteristic variables such as enterprise scale (*Sise*), leverage ratio (*Lev*), enterprise performance (*Roa*), board size (Board), listing years (*Year*) and macroeconomic variables such as industrial development level (*Inl*), financial development level (*Fin*), and opening level (*Open*) as control variables. Since the explanatory variable EPU is the annual time series data, to avoid the problem of multicollinearity, this paper does not control the year fixed effect in the model. Among them: enterprise scale (*Sise*) is represented by the natural logarithm of total assets, the leverage ratio (Lev) uses the ratio of the total debt to the total assets of the enterprise at the end of the period, reflecting the capital structure and solvency of the enterprise; corporate performance (Roa) is expressed by year-end net profit / annual average total assets; the size of the board of directors (Board) is expressed by the number of corporate board members; the number of years of listing (Year) indicates the length of the company's listing, and the difference in the number of years of listing may also have an impact on the investment efficiency of the company; The level of industrial development (Inl) is represented by the proportion of the added value of the secondary and tertiary industries in GDP; the level of financial development (Fin) is expressed by the proportion of RMB loan balances to GDP; the level of opening up (Open) is expressed by the proportion of import and export to GDP; the industry (Ind) is a dummy variable, and the value of the enterprise in the i industry is 1, otherwise the value is 0. In terms of robustness test, banking competition (Bank) and financing constraints (Sa) are introduced as control variables. Referring to the practice of Sheng et al. [25], this paper adopts the number of branches of five major state-owned industries/the number of branches of all commercial banks as the proxy variable of banking competition. The value is between 0 and 1, and the closer to 1, the lower the degree of banking competition. Financing constraints using Sa index.

**Mechanism variable.** Digital financial development (DFI). The digital inclusive financial index compiled by Guo et al. [14] is used as the proxy variable of digital financial development (DFI), and the natural logarithm is taken after arithmetic average of provincial data. Enterprise ESG performance (*ESG*). In this paper, Huazheng ESG evaluation index is used as the proxy

variable of enterprise ESG performance. The index divides the enterprise ESG performance into nine levels, from low to high, respectively C, CC, CCC, B, BB, BBB, A, AA, AAA. According to the practice of Gao et al. [26], the nine levels are assigned to 1–9 from low to high. Entrepreneur confidence (*Con*). This paper uses the entrepreneur confidence index compiled by the CSMAR database to measure entrepreneur confidence, Since the index is quarterly data, in order to unify it into annual data, we refer to the practice of Dong [5], and take the natural logarithm of the arithmetic mean of the four quarterly indicators each year as the annual indicator of the year.

**Heterogeneity variable.** Valuation level. In this paper, Tobin's Q value is used to measure the valuation of enterprises, which is expressed by the market value / total assets at the end of the year. This paper defines enterprises with Tobin's Q value greater than the median as high-valued enterprises, and vice versa as low-valued enterprises. Governance structure (*Top10*). Ownership concentration is closely related to corporate governance structure. This paper defines the top ten shareholders ' shareholding ratio exceeding the median as a shareholder-controlled enterprise with an assignment of 1, and vice versa as a manager-controlled enterprise with an assignment of 0. Audit reputation (*Big4*). If the enterprise cooperates with the four major international accounting firms, the enterprise is defined as a high audit reputation group with a value of 1, otherwise it is a low audit reputation group with a value of 0. Bank connection (*Bc*). Bank connection includes personnel connection (bank experience of senior executives or shareholders) and equity connection (bank equity held by enterprises). This paper refers to the practice of domestic related research [27, 28], and uses personnel connection to measure bank connection. Specifically defined as: if the company's executives or major shareholders now or have served in the bank, the value is 1, otherwise the value is 0. Table 2 shows the variable descriptions.

## 3.3. Data description

This paper selects the data of China's A-share listed companies from 2016 to 2021 as the research sample, and does the following processing: (1) Excluding financial industry

**Table 2. Variable descriptions.**

| Variable Symbols | Variable Name | Variable construction methods |
|---|---|---|
| Abs | corporate inefficient investment | The absolute value of the residual obtained by the ols estimation of Model (3) |
| Epu | economic policy uncertainty | The economic policy uncertainty index of Chinese mainland constructed by Huang Y. & Luk P. (2020) |
| Size | Enterprise size | The natural logarithm of the total assets of the enterprise at the end of the period |
| Lev | leverage ratio | Total end-of-period liabilities / total assets |
| Roa | enterprise performance | Net profit / total assets |
| Board | board size | Number of board of directors |
| Age | Years of listing of enterprises | Years of listing of enterprises |
| Inl | industrial development level | (Secondary industry added value + tertiary industry added value) / GDP |
| Fin | financial development level | RMB loan balances / GDP |
| Open | the level of opening | Total import and export / GDP |
| DFI | Development Level of Digital Finance | The natural logarithm is taken after the arithmetic average of the digital inclusive financial index. |
| ESG | Enterprise ESG performance | Huazheng ESG Evaluation Index |
| Con | entrepreneurial confidence | Entrepreneur confidence index compiled by CSMAR database |

**Table 3. Descriptive statistics.**

| Variables | Sample size | Average value | Standard deviation | Minimum value | Maximum value |
|---|---|---|---|---|---|
| *Abs* | 11022 | 0.0410 | 0.1208 | 0.0000 | 7.2467 |
| *Epu* | 11022 | 1.3791 | 0.0945 | 1.2840 | 1.5230 |
| *Size* | 11022 | 22.7075 | 1.3363 | 17.6500 | 28.6400 |
| *Lev* | 11022 | 0.4499 | 0.2004 | 0.0084 | 1.6981 |
| *Roa* | 11022 | 0.0304 | 0.0783 | -1.6479 | 0.7859 |
| *Board* | 11022 | 8.5666 | 1.7054 | 0.0000 | 17.0000 |
| *Age* | 11022 | 14.2387 | 6.8404 | 2.0000 | 31.0000 |
| *Inl* | 11022 | 0.9255 | 0.0035 | 0.9194 | 0.9296 |
| *Fin* | 11022 | 1.5493 | 0.1100 | 1.4283 | 1.7043 |
| *Open* | 11022 | 0.3286 | 0.0083 | 0.3179 | 0.3418 |
| *DFI* | 11022 | 3.0671 | 0.4638 | 2.3041 | 3.7272 |
| *ESG* | 11022 | 6.5734 | 1.2276 | 1.0000 | 9.0000 |
| *Con* | 11022 | 4.8091 | 0.0094 | 4.7950 | 4.8195 |

companies; (2) Excluding ST and ST* companies; (3) Remove companies with missing data. Finally, 1837 companies and 11022 sample values were obtained. The economic policy uncertainty (EPU) index uses the economic policy uncertainty index of mainland China constructed by Huang Y. & Luk P. [24]. The ESG performance data of enterprises are selected from the Huazheng ESG index, and the remaining data are from the CSMAR database. In this paper, stata17.0 software was used for statistical analysis. Table 3 shows the descriptive statistics.

## 4. Empirical analysis

### 4.1. Benchmark regression results

It can be seen from Column (1) of Table 4 that without controlling any variables, the influence coefficient of economic policy uncertainty (EPU) on inefficient investment of enterprises is 0.0891, which is significant at the level of 1%. It can be seen from column (2) that the influence coefficient is 0.0714, and it is significant at the level of 1% when only micro variables are controlled. It can be seen from Column (3) that the influence coefficient is 0.113 and the significance level is 1% when the micro and macro variables are controlled at the same time. In summary, EPU has a significant positive impact on corporate inefficient investment, that is, it has a significant negative impact on corporate investment efficiency. Therefore, the previous hypothesis H1 is established. The uncertainty of economic policy during the sample period is high. On the one hand, this makes enterprises face a more complex investment environment, and managers or decision makers look ahead, give up or miss investment opportunities; on the other hand, this makes credit constraints and financing more difficult, and enterprises have to pay more investment costs even if they have the willingness to invest, resulting in a decline in the investment efficiency of enterprises.

### 4.2. Robustness test

Because the economic policy uncertainty index is monthly data, this paper refers to the method of Li et al. [29], and takes the data of the twelfth month of each year (EPU-dec) as the proxy variable of EPU for regression. In addition, this paper also uses the fiscal policy uncertainty index (FEPU) and monetary policy uncertainty index (MEPU) of mainland China developed by Huang Y. & Luk P. [24] as new explanatory variables for regression. The results are significantly positive as shown in column (3), column (4) and column (5) of Table 5. Then, the

**Table 4. Baseline regression results.**

| Explanatory Variable | Explained variable | | |
|---|---|---|---|
| | (1) | (2) | (3) |
| EPU | 0.0891*** | 0.0714*** | 0.1129*** |
| | (5.94) | (4.97) | (2.59) |
| Size | | 0.0202* | 0.0209* |
| | | (1.76) | (1.81) |
| Lev | | 0.0908** | 0.0878** |
| | | (2.37) | (2.30) |
| Roa | | 0.0840*** | 0.0784*** |
| | | (4.14) | (3.92) |
| Board | | -0.0026 | -0.0026 |
| | | (-1.42) | (-1.42) |
| Age | | -0.0093*** | 0.0091 |
| | | (-6.88) | (1.60) |
| Inl | | | -2.5961** |
| | | | (-2.40) |
| Fin | | | -0.2627*** |
| | | | (-2.79) |
| Open | | | 0.506* |
| | | | (1.78) |
| _cons | -0.0819*** | -0.4040 | 1.905* |
| | (-3.96) | (-1.57) | (1.82) |
| Individual fixed effects | yes | yes | yes |
| N | 11022 | 11022 | 11022 |

Notes: The robustness standard errors, *, **, *** in brackets represent the significance levels of 10%, 5% and 1%, respectively. These are consistent in the following tables.

control variables are adjusted. Firstly, the control variables *Roa* and *Board* are replaced by return on equity (Roe), duality (Dua) and the proportion of independent directors (Pid). In addition, in view of the fact that China's micro-enterprise financing is still dominated by bank credit, this paper increases the two control variables of banking competition (Bank) and financing constraints (*Sa*) for regression. The test results are shown in columns (1) and (2) of Table 5. After replacing and increasing the control variables, the conclusions are consistent.

## 4.3. Endogeneity test

The IV can alleviate the problem of missing variables and reverse causality to a certain extent. Therefore, this paper uses the IV method to estimate. Since the US economic policy is an important factor affecting the formulation and implementation of China's economic policy, and the uncertainty of the US economic policy will affect the behavior of Chinese enterprises by affecting the uncertainty of China's economic policy [29]. Therefore, this paper draws on the practice of Li et al. [29], and selects the U.S. Economic Policy Uncertainty Index (*UEPU*) developed by Baker et al. [22] as an exogenous instrumental variable of China's Economic Policy Uncertainty (*Epu*) for testing. Of course, this indicator is also unified into an annual indicator in the same way as the *EPU*. To this end, we add Model (4) on the basis of Model (1), and use the two models for IV estimation:

$$UEPU_t = \gamma_0 + \gamma_1 Epu_t + \gamma_2 X_{it} + \eta_i + \varepsilon_{it} \tag{4}$$

**Table 5. Robustness test results.**

| Explanatory Variable | (1) | (2) | (3) | (4) | (5) |
|---|---|---|---|---|---|
| | Replace control variables | | Increase control variables | | Replace explanatory variables |
| EPU | 0.105** | 0.112*** | | | |
| | (2.33) | (2.65) | | | |
| EPU-dec | | | 0.0118*** | | |
| | | | (2.59) | | |
| FPU | | | | 0.0347*** | |
| | | | | (2.59) | |
| MPU | | | | | 0.0551*** |
| | | | | | (2.59) |
| Control variables | Yes | Yes | Yes | Yes | Yes |
| Individual fixed effects | Yes | Yes | Yes | Yes | Yes |
| N | 11022 | 11022 | 11022 | 11022 | 11022 |

Table 6 reports the IV estimation results of the two-stage least squares method. Among them, (1) is listed as the first stage regression, and the results show that the US economic policy uncertainty (*UEPU*) has a significant negative impact on China's economic policy uncertainty (*EPU*), which meets the correlation requirements of instrumental variables; column (2) is the second-stage regression result. EPU has a significant positive impact on the inefficient investment of enterprises, which is consistent with the previous conclusions. The *Kleibergen-Paap rk* Wald F value is 9518.18, which is much larger than the *Stock-Yogo* threshold at the 10% level, indicating that the risk of weak instrumental variables is very small, we chose the appropriate instrumental variables. The results of Table 6 show that our research results are still valid after using the instrumental variable method to deal with the endogeneity problem of the model.

## 4.4. Mechanism test results

As shown in column (1) of Table 7, the influence coefficient of the cross-multiplication term of EPU and digital financial development on the non-investment efficiency of enterprises is -0.182, and it is significant at the level of 5%, indicating that the development of digital finance weakens the impact of EPU on the investment efficiency of enterprises, assuming that H2 is verified. The reasons for the development of digital finance to exert this regulatory effect may be as follows: first, to alleviate the problem of information asymmetry. On the one hand, the development of digital finance can strengthen the connection between banks and enterprises

**Table 6. Two-stage estimation results of IV.**

| Explanatory Variable | EPU | Abs |
|---|---|---|
| | (1) | (2) |
| EPU | | 0.1913*** |
| | | (2.91) |
| UEPU | -0.0854*** | |
| | (-97.56) | |
| Kleibergen-Paap rk Wald F value | | 9518.18 |
| Stock-Yogo critical value (10%) | | 16.38 |
| Control variables | yes | yes |
| Individual fixed effects | yes | yes |
| N | 11022 | 11022 |

**Table 7. Mechanism test results.**

| Explanatory Variable | Explained variable | | |
|---|---|---|---|
| | (1) | (2) | (3) |
| EPU | 0.7648** (2.46) | 0.7648** (2.46) | 0.1472*** (4.18) |
| DFI | 0.4278 (1.34) | | |
| EPU*DFI | -0.182** (-2.42) | | |
| ESG | | 0.0364** (3.21) | |
| EPU*ESG | | -0.0276*** (-3.19) | |
| CON | | | 0.155*** (4.13) |
| EPU*CON | | | -0.119*** (-4.20) |
| Control variables | Yes | Yes | Yes |
| Individual fixed effects | Yes | Yes | Yes |
| N | 11022 | 11022 | 11022 |

and reduce the cost of enterprise information search. On the other hand, it can promote financial institutions to better grasp enterprise operation and risk information and make credit decisions in time. Second, broaden the financing channels of enterprises. Under the traditional financial model, the financing channels of enterprises are relatively single, and the development of digital finance can broaden the financing channels of enterprises, improve the convenience and coverage of financial services, and improve the financing efficiency of enterprises.

As shown in Column (2) of Table 7, the cross term of EPU and enterprise ESG performance has a negative impact on the inefficient investment of enterprises, with a coefficient of -0.0276 and passes the 1% statistical significance test. It shows that strong corporate ESG performance will reduce the negative impact of economic policy uncertainty on corporate investment efficiency, assuming that H3 is verified. The possible economic explanation is as follows: First, revise the corporate investment strategy. When faced with economic policy uncertainty, managers with political connections or information resources may adopt radical investment strategies, resulting in overinvestment; most enterprise managers will invest cautiously, resulting in insufficient investment. Strong ESG performance will enable business managers to make more reasonable investment decisions based on long-term vision, thereby correcting the deviation of corporate investment decisions caused by economic policy uncertainty and improving corporate investment efficiency. Second, play the information effect. Strong ESG performance will improve the quality of information disclosure of enterprises and send positive signals to external investors and financial institutions.

As shown in Column (3) of Table 7, the interaction term of economic policy uncertainty and entrepreneurial confidence has a negative impact on the inefficient investment of enterprises, and is significant at the level of 1%. It shows that entrepreneurial confidence will play a negative regulatory role in the process of EPU affecting the investment efficiency of enterprises, assuming that H4 is verified. The economic reason for the moderating effect of entrepreneurial confidence may be that entrepreneurial confidence will promote managers to more effectively examine the internal situation and external policy environment of enterprises, so as to make reasonable expectations and decisions and improve investment efficiency.

## 4.5. Heterogeneity analysis

Based on the previous research, we will focus on the heterogeneous impact of economic policy uncertainty on different types of enterprises. Of course, this heterogeneity analysis is mainly carried out in the context of placing enterprises in the context of economic policy fluctuations. The first is the enterprise valuation. In the face of drastic fluctuations in economic policy, the difference in valuation level may make the investment decisions of enterprises have a greater difference. The second is the governance structure. The difference in governance structure will also lead to different coping strategies for enterprises to deal with economic policy uncertainty. Obviously, the risk appetite and interest demand of shareholder-controlled enterprises and manager-controlled enterprises are different. The third is audit reputation. Considering that the financial management of enterprises will have a significant impact on corporate investment, it is necessary to study the heterogeneity of enterprise samples with different audit reputations. The fourth is bank-enterprise linkage. Since bank loans are still the main source of funds for Chinese enterprises, we believe that it is necessary to examine the differences in investment efficiency of different enterprises in response to uncertainty shocks from the perspective of bank-enterprise linkage.

Tobin's q value is widely used in the measurement of enterprise value. According to the size of Tobin's q value, this paper divides the sample enterprises into high valuation enterprises and low valuation enterprises. Table 8; as shown in columns (1) and (2), in the low valuation group, the coefficient of the impact of EPU on the non-investment efficiency of enterprises is 0.144, and it is significant at the level of 1%; in the high valuation group, the results were not significant. Compared with high-valuation enterprises, EPU has a greater impact on the investment efficiency of low-valuation enterprises. According to the research of Gu & Zhu [8]: on the one hand, because the Tobin Q value reflects the future development trend of the enterprise, there are many investment opportunities, and the enterprise will carefully examine the investment plan and pay attention to the improvement of investment efficiency. On the other hand, the high valuation of enterprises indicates that investors in the market have high recognition of enterprises and low degree of information asymmetry. When faced with financing constraints, it is easier for enterprises to raise funds and alleviate underinvestment.

The shareholding ratio of the top ten shareholders can better reflect the corporate governance structure. In this paper, the sample enterprises are divided into shareholder enterprises and manager enterprises according to the shareholding ratio of the top ten shareholders. As shown in column (3) (4) of Table 8, in the shareholder control group, the influence coefficient of EPU on non-investment efficiency of enterprises is 0.138, and it is significant at the level of 5%; in the manager control group, the results are not significant. It shows that compared with managers-controlled enterprises, the investment efficiency of shareholders-controlled

**Table 8. Results of heterogeneity test.**

| Explanatory Variable | High-valuation | Low-valuation | shareholder type | Manager type | High-audit reputation | Low-audit reputation | bank-related group | non-bank-related group |
|---|---|---|---|---|---|---|---|---|
| | **(1)** | **(2)** | **(3)** | **(4)** | **(5)** | **(6)** | **(7)** | **(8)** |
| *EPU* | 0.144*** | 0.0822 | 0.138** | 0.102 | 0.141 | 0.103** | 0.164 | 0.111** |
| | (2.71) | (0.74) | (2.11) | (1.59) | (0.84) | (2.29) | (0.96) | (2.5) |
| *Control variables* | Yes | Yes | Yes | Yes | Yes | Yes | Yes | Yes |
| *Individual fixed effects* | Yes | Yes | Yes | Yes | Yes | Yes | Yes | Yes |
| *N* | 5511 | 5511 | 5509 | 5513 | 780 | 10242 | 156 | 10866 |

enterprises is more significantly affected by economic policy uncertainty. The equity of shareholder-controlled enterprises is highly concentrated, and the major shareholders control the absolute control of the enterprise. They often over-intervene in the management and decision-making of the enterprise, making the enterprise make more irrational investment decisions in the face of high economic policy uncertainty. Reduce investment efficiency [8]. Manager-controlled enterprises have a higher degree of separation of two rights. Professional managers ' more professional quality and more rational decision-making can improve governance efficiency and investment efficiency.

As shown in columns (5) and (6) of Table 8, in the low audit reputation group, the influence coefficient of EPU on enterprise investment efficiency is 0.103, and it is significant at the level of 5%. On the contrary, it is not significant in the high audit reputation group. This shows that compared with the high audit reputation group, economic policy uncertainty has a greater impact on the investment efficiency of enterprises with low audit reputation. According to the research of Gu & Zhu [8]: the possible explanation is that the quality of accounting report information of enterprises with low audit reputation is low. In the case of economic policy uncertainty, investors reduce their investment willingness and investment scale to avoid information asymmetry, which aggravates the financing constraints of enterprises and reduces the investment efficiency of enterprises.

Under the current situation of bank credit as the main financing method, bank-enterprise connection (especially personnel connection) will significantly affect the investment efficiency of enterprises. In this paper, according to the criteria that senior executives or major shareholders now or have served in banks, the sample enterprises are divided into bank-related groups and non-bank-related groups. As shown in columns (7) and (8) of Table 8, in the non-bank-related group, EPU has a significant positive impact on the inefficient investment of enterprises. In the group with bank connections, the influence coefficient of EPU on enterprise investment efficiency is 0.164, but not significant. This shows that EPU has a greater impact on the investment efficiency of non-bank affiliated enterprises than that of bank affiliated enterprises. According to the research of Zhai et al. [30]: The possible explanation is that enterprises with bank connections have better financing advantages and resources, which can effectively alleviate the financing constraints caused by EPU to enterprises, so as to maintain higher investment efficiency. The financing cost of enterprises without bank connection is higher and the financing is more difficult, so it is difficult to eliminate the impact of EPU on investment efficiency.

## 5. Conclusions

Based on the data of China's A-share listed companies from 2016 to 2021, We construct a panel fixed effect model, and conducts empirical research and theoretical explanation on the linear relationship, mechanism and heterogeneity characteristics of macroeconomic policy uncertainty and micro-enterprise investment efficiency. The research conclusions are as follows: First, EPU has a significant negative impact on corporate investment efficiency. The increase of EPU will reduce the investment efficiency of enterprises. The above conclusions are still robust after replacing the explanatory variables, adjusting the control variables, and estimating the propensity score matching. In fact, many scholars in other countries have obtained similar research results on the impact of economic policy uncertainty on corporate entities, in addition to the above conclusions drawn by us with China as the research object. For example, Jens [31] highlights the effect of the presidential election on firm investments, showing that investments decrease before and rebound after an election. According to Kahle & Stulz [32], capital expenditures and corporate borrowing usually fall sharply in times of

policy uncertainties and financial crises. Gulen & Ion [16] use the EPU index to argue the impact of uncertainty on corporate decisions. Secondly, the results of mechanism analysis show that digital financial development, enterprise ESG performance and entrepreneur confidence can negatively regulate the impact of economic policy uncertainty on enterprise investment efficiency and reduce the decline of enterprise investment efficiency. Thirdly, the results of heterogeneity analysis show that the impact of economic policy uncertainty on the investment efficiency of different types of enterprises has significant heterogeneity characteristics. Specifically, compared with enterprises with high valuation, manager control, high audit reputation and bank connection, EPU has a greater negative impact on the investment efficiency of enterprises with low valuation, shareholder control, low audit reputation and no bank connection.

In addition, our findings also have the following policy implications: 5.1 Improve the stability and pertinence of economic policies. First, we should pay more attention to consistency and coherence in policy formulation, pay more attention to transparency and timeliness in policy release, and give micro-subjects stable policy expectations. Second, it is necessary to establish an in-depth policy effect evaluation and dynamic adjustment mechanism to improve the effectiveness of economic policies. Third, we should optimize the business environment, actively guide public opinion, respect entrepreneurship, and encourage entrepreneurs to innovate. 5.2 Vigorously promote the development of digital finance. First, it is necessary to broaden the use of digital finance, optimize the digital financial market mechanism, expand the coverage of digital financial services, and ease corporate financing constraints. Second, it is necessary to speed up the iterative upgrading of digital financial technology and guide the rational and efficient flow of key elements such as digital financial technology and digital financial talents. Third, it is necessary to encourage the extensive use and innovative application of digital technology by micro-enterprises and financial institutions to alleviate the information asymmetry and moral hazard problems in the credit market. 5.3 Improve the corporate governance structure. First, we should actively adjust the corporate governance system, improve the quality of internal control, balance ownership and management rights, and improve the shareholder restraint mechanism. The second is to strengthen the construction of enterprise ESG, actively assume ESG responsibilities, scientifically formulate ESG strategic planning, and improve the quality of ESG information disclosure. The third is to improve the quality of accounting information, improve the comprehensive quality of enterprise financial management personnel, strengthen the supervision of accounting basic work, and enhance the audit reputation of the enterprise itself. Fourth, moderately strengthen the relationship between banks and enterprises. On the one hand, it is possible to strengthen equity cooperation with financial structures such as commercial banks to strengthen equity linkages; on the other hand, we should actively introduce management talents with bank background and strengthen the personnel connection with banks.

## Supporting information

**S1 Data.**
(XLSX)

## Author Contributions

**Conceptualization:** Bing Zhou, Licheng Bie.

**Writing – review & editing:** Bo Liang.

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
