## [Decision Letter · Decision Letter 0]

12 Mar 2024

PONE-D-24-06675The impact of macroeconomic policy uncertainty on micro-enterprise investment efficiency—Empirical evidence from ChinaPLOS ONE

Dear Dr. Bo Liang,

Thank you for submitting your manuscript to PLOS ONE. After careful consideration, we feel that it has merit but does not fully meet PLOS ONE’s publication criteria as it currently stands. Therefore, we invite you to submit a revised version of the manuscript that addresses the points raised during the review process.

We look forward to receiving your revised manuscript.

Kind regards,

Ricky Chee Jiun Chia

Academic Editor

PLOS ONE

Journal Requirements:

3. PLOS requires an ORCID iD for the corresponding author in Editorial Manager on papers submitted after December 6th, 2016. Please ensure that you have an ORCID iD and that it is validated in Editorial Manager. To do this, go to ‘Update my Information’ (in the upper left-hand corner of the main menu), and click on the Fetch/Validate link next to the ORCID field. This will take you to the ORCID site and allow you to create a new iD or authenticate a pre-existing iD in Editorial Manager. Please see the following video for instructions on linking an ORCID iD to your Editorial Manager account: https://www.youtube.com/watch?v=_xcclfuvtxQ.

4. We note that your Data Availability Statement is currently as follows: [All relevant data are within the manuscript and its Supporting Information files]

Reviewers' comments:

Reviewer's Responses to Questions

**Comments to the Author**

1. Is the manuscript technically sound, and do the data support the conclusions?

Reviewer #1: Partly

Reviewer #2: Partly

2. Has the statistical analysis been performed appropriately and rigorously? 

Reviewer #1: Yes

Reviewer #2: Yes

3. Have the authors made all data underlying the findings in their manuscript fully available?

Reviewer #1: Yes

Reviewer #2: Yes

4. Is the manuscript presented in an intelligible fashion and written in standard English?

Reviewer #1: Yes

Reviewer #2: No

5. Review Comments to the Author

Reviewer #1: The analysis of the impact of macroeconomic policy uncertainty on micro-enterprise investment efficiency in the article yielded meaningful conclusions. However, it is important to note that there are deficiencies in various sections of the article. Therefore, my review conclusion is that a Major Revision is required. I hope the authors can improve the article based on the following suggestions, although these revisions may not be easy to implement:

1.The introduction section is incomplete, as it fails to clearly identify the gap in existing research and the importance of this study.

2.The rationale for focusing on China as the study's subject, especially considering PLOS is an international journal, needs to be clearly explained. Additionally, the paper lacks a comparison of its findings with those from other countries, a comparison that should be present in the introduction, literature review, or conclusion sections.

3.I have doubts about the key explanatory variable. Macroeconomic policy is multifaceted, so is it reasonable to measure it using only one index? Additionally, how is it measured and calculated specifically? Merely citing literature does not give me sufficient understanding of this index. Finally, have any journals published research using this index? Similarly, the selection of the key explained variable still needs literature support, as this indicator is quite indirect, as mentioned in the article.

4.The discussion and resolution of endogeneity are insufficient. First, the specifics of the PSM operation are unclear, such as what matching methods were used and other more detailed aspects of the matching process? Second, for reporting PSM results, it is recommended to regress using the matched sample and compare it with the baseline regression results. Third, I strongly suggest the authors try using instrumental variable methods to address potential endogeneity issues (caused by reverse causality or omitted variables, etc.).

5.Although the model setup for mechanism analysis is stated, Table 6 still needs to show the coefficients and significance levels of the mechanism variables (DFI, ESG & CON).

6.Any heterogeneity analysis must have a theoretical basis, i.e., why such heterogeneity analysis is conducted. Additionally, the explanation of the results of the heterogeneity analysis part needs support from relevant literature.

7.From the results of the entire text, it is confusing which time dimension the regression is conducted on. For example, in the model setup section, the author mentioned the time dimension is on an annual basis (year basis). However, some key variables are measured on a monthly and quarterly basis, and it is not detailed how the final indicators are calculated or summed to a fixed time dimension.

8.The conclusion section is very weak; merely summarizing the results of the entire text is insufficient. I suggest adding some necessary discussions, which could be policy implications or comparisons with the conclusions of existing related research.

Hope you everything well.

Reviewer #2: 1. The English writing should be improved.

2. Could you provide the results of sample balance test ?

3. Do you consider IV to deal with endogeneity problem.

4. Do you consider the unobserved factors varying with time?

6. PLOS authors have the option to publish the peer review history of their article (what does this mean?). If published, this will include your full peer review and any attached files.

Reviewer #1: No

Reviewer #2: No

---

## [Author Response · Author response to Decision Letter 0]

7 May 2024

We would like to express our heartfelt gratitude to two anonymous reviewers for their valuable feedback. Our manuscript, titled “The impact of macroeconomic policy uncertainty on micro-enterprise investment efficiency—Empirical evidence from China”, benefited significantly from the constructive comments and insights from the review team. Based on the suggestions received, we have made careful revisions to the original manuscript. In the revised manuscript, all changes are marked in red. In addition, we have also carefully proofread this manuscript for typographical, grammatical, and other errors. We hope the revised manuscript can meet your standard of quality and address the concerns raised by the reviewers.

---

## [Editor Report · Decision Letter 1]

16 May 2024

The impact of macroeconomic policy uncertainty on micro-enterprise investment efficiency—Empirical evidence from China

PONE-D-24-06675R1

Dear Dr. Bo Liang,

We’re pleased to inform you that your manuscript has been judged scientifically suitable for publication and will be formally accepted for publication once it meets all outstanding technical requirements.

Kind regards,

Ricky Chee Jiun Chia

Academic Editor

PLOS ONE
---

## [Editor Report · Acceptance letter]

2 Jun 2024

PONE-D-24-06675R1 

PLOS ONE

Dear Dr. liang, 

I'm pleased to inform you that your manuscript has been deemed suitable for publication in PLOS ONE. Congratulations! Your manuscript is now being handed over to our production team.

Kind regards, 

on behalf of

Dr. Ricky Chee Jiun Chia 

Academic Editor

PLOS ONE